# Composite Hemostatic Nonwoven Textiles Based on Hyaluronic Acid, Cellulose, and Etamsylate

**DOI:** 10.3390/ma13071627

**Published:** 2020-04-01

**Authors:** Pavel Suchý, Alice Paprskářová, Marta Chalupová, Lucie Marholdová, Kristina Nešporová, Jarmila Klusáková, Gabriela Kuzmínová, Michal Hendrych, Vladimír Velebný

**Affiliations:** 1Department of Human Pharmacology and Toxicology, Faculty of Pharmacy, University of Veterinary and Pharmaceutical Sciences, Brno 612 42, Czech Republic; 2Contipro a.s., Dolní Dobrouč 561 02, Czech Republic; 3Department of Physiology, Faculty of Medicine, Masaryk University, Brno 625 00, Czech Republic; 4First Department of Pathology, St. Anne’s University Hospital and Faculty of Medicine, Masaryk University, Brno 656 91, Czech Republic

**Keywords:** hemostasis, hyaluronic acid, cellulose, etamsylate, nonwoven textile

## Abstract

The achievement of rapid hemostasis represents a long-term trend in hemostatic research. Specifically, composite materials are now the focus of attention, based on the given issues and required properties. In urology, different materials are used to achieve fast and effective hemostasis. Additionally, it is desirable to exert a positive influence on local tissue reaction. In this study, three nonwoven textiles prepared by a wet spinning method and based on a combination of hyaluronic acid with either oxidized cellulose or carboxymethyl cellulose, along with the addition of etamsylate, were introduced and assessed in vivo using the rat partial nephrectomy model. A significantly shorter time to hemostasis in seconds (*p* < 0.05), was attributed to the effect of the carboxymethyl cellulose material. The addition of etamsylate did not noticeably contribute to further hemostasis, but its application strengthened the structure and therefore significantly improved the effect on local changes, while also facilitating any manipulation by the surgeons. Specifically, the hyaluronic acid supported the tissue healing and regeneration, and ensured the favorable results of the histological analysis. Moreover, the prepared textiles proved their bioresorbability after a three-day period. In brief, the fabrics yielded favorable hemostatic activity, bioresorbability, non-irritability, and had a beneficial effect on the tissue repair.

## 1. Introduction

Hemostasis represents a dynamic and complex multistage system. It is a physiological process that protects the organism from excessive blood loss. Hemostasis is provided in several stages using three mechanisms: primary hemostasis, secondary hemostasis, and fibrinolysis. In these processes, vascular wall reaction, thrombocyte activity, clotting cascade activation, and the activity of coagulation factors with fibrin formation and fibrinolysis are involved [1]. If the body’s own mechanisms are not effective enough to stop the bleeding, it is necessary to apply additional hemostatic agents. Currently, surgeons use different methods to achieve rapid hemostasis. Physical methods such as manual compression and suturing may not be completely effective. Thermal methods such as laser, bipolar coagulation, and ultrasonic dissectors may cause serious complications connected with the development of infection and necrosis formation [2,3]. In human medicine, there are a number of different materials designed for efficient hemostasis. These materials differ from each other in composition, form, mechanism of action, and limitations of application. Recent trends focus on fast and effective hemostatic action together with a positive contribution to healing and tissue recovery. Difficult-to-handle, non-bioresorbable materials associated with irritability are very common [4,5]. The removal of the applied preparation once hemostasis is achieved is often required. Unfortunately, any further intervention in the function of the body carries a potential risk. A wide range of traditional materials based on cellulose, collagen, and gelatine are used for varying medical applications [6], but there are now many new fabrics being introduced, including modified sodium starch glycolate, a new chitosan hydroquinone-based gauze, hemostatic agents based on a gelatin-microbial transglutaminase mix, along with other materials with mineral content or metal ion-chelated tannic acid coating [7,8,9,10,11]. Different composites, such as oxycellulose cross-linked gelatin microparticles, a plant-derived oxidized nanofibrillar cellulose-chitosan hemostat and thrombin with a gelatin sponge carrier, have appeared on the market as well [12,13,14].

The new fabrics introduced in this study are based on the utilization of hyaluronic acid, a linear polysaccharide composed of disaccharide units composed of glucuronic acid linked with N-acetyl-glucosamine using a β-1,3 glycosidic bond. Single subunits are connected together via a β-1,4 bond. The substance is naturally present in the human body and participates in many biological processes. It is neither immunogenic nor toxic; therefore, it is suitable for use in various medical applications [15]. Hyaluronic acid is widely used in tissue engineering and regenerative medicine. It has gained a strong position, especially in the healing of wounds and as a scaffold for different biologically-active substances [16,17,18]. Besides its healing effect, its hemostatic potential has also been widely discussed [19]. Furthermore, all the prepared materials were partly formed from cellulose, which is a linear polymer consisting of D-glucose linked by β-glycosidic bonds. Cellulose materials of different structure have also gained an important position in healing wounds and hemostatic intervention. Carboxymethyl cellulose is a derivative of cellulose modified by carboxymethyl groups (-CH2-COOH), which are bound to hydroxyl groups. The base of the structure is made up of glucopyranose monomers. Commercially available preparations contain carboxymethyl cellulose with varying degrees of substitution (DS) [20]. DS represents an important factor that can influence hemostatic properties, depending on the absorption capacity and dissolution of the prepared material. Carboxymethyl cellulose is widely used in the form of sodium salt, which is well tolerated and soluble in water. It is applied primarily as a wound-healing dressing or as an implantable material. Upon closer examination, the hemostatic potential has also been revealed [21,22]. So far, composite materials utilizing hyaluronic acid and carboxymethyl cellulose have been developed as a barrier against undesired adhesion in surgery [23]. Oxidized cellulose, another derivative of cellulose, is a biocompatible polymer. The base structure of cellulose is modified by the partial oxidation of the hydroxymethyl groups on the hydroglycose rings. This process of oxidation, typically performed by dinitrogen tetroxide, promotes the hemostatic properties of fibers and the susceptibility to degradation by glycosidases. It is possible to prepare two types of this derivative: oxidized regenerated cellulose, characterized by the formation of organized fibers prior to oxidation, and oxidized non-regenerated cellulose [24]. Hemostats using oxidized cellulose have become very popular due to their efficacy and ease of use. Moreover, oxidized cellulose also possesses antibacterial properties [25]. Composite materials based on hyaluronic acid and oxidized cellulose were also employed in the prevention of postsurgical adhesion [26]. Furthermore, etamsylate was used in the development process. It is an active pharmaceutical substance with a hemostatic effect. According to its chemical structure, it is a 2,5-dihydroxy benzene sulphonic acid with diethylamine. Etamsylate is a unique synthetic drug with antihemorrhagic properties, which influences the first step of hemostasis. Specifically, it improves the adhesiveness of platelets and restores capillary resistance. However, the complete mechanics of these actions remains unknown. Its therapeutic benefits have branched out to many surgical specialties. The agent is commonly used to stop bleeding or to prevent hemorrhage by local or systemic application [27,28]. Efforts to apply this substance on a substrate have been reported previously [29]. The results of our tests imply that the composition of our substrate is unique for its hemostatic and healing properties.

Recent trends in hemostatic research focus on composite materials with the aim of achieving fast and efficient hemostasis [30]. In urology, quick hemostasis is very desirable in instances of uncontrollable bleeding [31,32]. Moreover, advanced wound healing, based on tissue regeneration and restoration of its function, are now the center of interest [33].

The aim of this study was to prepare a unique bioresorbable material with strong hemostatic and healing properties and then to evaluate its effect in comparison with the reference material, a regenerated oxidized cellulose textile (ORC). Therefore, new textiles based on hyaluronic acid and various cellulose derivatives have been developed. This study also introduces a material that combines an active biological effect with the biopolymer’s physical mechanics. Finally, three different preparations of a combined composition have been introduced in the study: a composite material of hyaluronic acid and oxidized cellulose (HA/OX), a second with hyaluronic acid and carboxymethyl cellulose (HA/CMC), and a third made of hyaluronic acid and carboxymethyl cellulose with the addition of etamsylate (HA/CMC/ET). The materials were tested using a rat partial-nephrectomy model, which represents a simple but appropriate, useful, quality model, and one which has yielded valid results.

## 2. Materials and Methods

### 2.1. Materials

Hyaluronic acid (HA; molecular weight 990 g·mol^−1^; polydispersity 1.772) was obtained from Contipro a.s. (Dolní Dobrouč, Czech republic). Carboxymethyl cellulose (CMC) in the form of sodium salt (degree of substitution 0.7, molecular weight 250 g·mol^−1^) was supplied by Sigma Aldrich, while oxycellulose (OX) in the form of calcium salt (Okcel Ca-L powder; carboxyl groups content 17.28%) came from Synthesia (Pardubice, Czech Republic). Etamsylate (ET; molecular weight 263.31 g·mol^−1^) was obtained from TCI Europe NV. Reference oxidized regenerated cellulose (ORC; absorbable hemostat Pahacel^®^fibril) is from Pahacel, Altaylar Medical (Sincan, Turkey). The following chemicals were used for gene expression analysis: RNAzol^®^ RT (Sigma Aldrich, St. Louis, MO, USA), RNeasy mini kit (QIAGEN, Hilden, Germany), High-Capacity Reverse Transcription kit (Thermo Fisher Scientific, Waltham, MA, USA), DEPC-treated water, and TaqManTM qPCR assays (Thermo Fisher Scientific): RPL13A Rn00821946_g1, TGFB1 Rn00572010_m1, TNF Rn00562055_m1.

### 2.2. Preparation of Hemostatic Textiles

The study represents three new hemostatic materials. The basic structure was prepared with the use of the wet spinning method in a coagulation bath comprised of 2-propanol. The fibers were obtained by precipitation in the bath. The entire process was derived from the patented procedures of Contipro a.s. [34,35,36]. The workflow is illustrated using a schematic diagram (Figure 1).

For the preparation of a 2% mixture of hyaluronic acid and cellulose, both substances were mixed together in one bottle and filled with distilled water. The mixture was left for 16–24 h in an incubator shaker, Innova 40/40 R (Eppendorf Czech & Slovakia s.r.o., Říčany, Czech Republic), at 37 °C, 200 rpm. The prepared mixture can be refrigerated (at 2–8 °C), although it had to be warmed up to room temperature and re-homogenized before using. Initially, a ratio of 1:1 for each textile was used. Eventually, the aim was to increase the amount of cellulose in the composite material to achieve the greatest hemostatic potential. The weight ratio of 3:7 was found to be the limit in preparing the HA/CMC nonwoven textile with acceptable technological properties, which were examined using the bursting strength tester. The working solution was clear and slightly viscose with a yellowish color. For the preparation of the second mixture, oxidized cellulose calcium salt was suspended in a solution of hyaluronic acid. The weight ratio was 1:1. Calcium salt of oxycellulose, unlike sodium salt of carboxymethyl cellulose, is insoluble in water. In this case the ratio of 1:1 was the limit. It was not possible to increase the amount of oxycellulose and prepare the textile. The suspension was white and slightly viscose, with a cloudy appearance.

The prepared suspension/solution was then processed by the above-mentioned wet spinning method using the company’s exclusive spinning laboratory equipment. The principle of the fiber’s creation was precipitation of the polymeric mixture in the flow of the non-stationary coagulation bath with the 2-propanol content. The mixture was then dosed by a peristaltic pump through spinning tubes into the flow of bath with the speed set at 10 mL/min. An extrusion nozzle designed for dosing was located vertically in the direction of the bath flow. The flow was propelled continuously by a connected pump. The precipitation of the mixture resulted in the creation of soft, thin fibers. The new fibers floating in the stream were then captured by system of comb separators. The fibers were then transferred into the beaker with 2-propanol, using combs and smooth tweezers, to cure and increase their strength.

The created fibers were cut with rotational knives in a container with 2-propanol for approximately 5 s at 12,000 rpm (MICROTRON^®^ MB 800, Kinematica, Luzern, Switzerland). The mixture of shortened fiber was then filtered through a porous substrate made of polyamide knitwear, which was supported by an iron frame. For the experiment, an 8 × 8 cm form was used. The obtained layer of fibers on the substrate was then transferred into a drying machine created for laboratory experiments to remove the coagulant. Drying was performed at 50 °C for 30 min.

Due to the spinning efficiency of both components, the first final product contained hyaluronic acid and oxidized cellulose at a ratio of 1:1, and the second contained hyaluronic acid and carboxymethyl cellulose at a ratio of 1:3. The results were obtained by nitrogen content analysis (method f.Leco, EKO-LAB Žamberk, Žamberk, Czech Republic). Specific areal weight was achieved using the exact amount of the mixture, with the areal weight of 50 g·m^−2^ being defined as the most appropriate, considering the required effect and the desirable handling properties during surgery. Therefore, 19.5 mL of the HA/OX mixture and 20.0 mL of the HA/CMC mixture were used to prepare the required quantity.

### 2.3. Modification of the Prepared Textile

Creating a composite material with the greatest hemostatic potential was the desired goal. The amount of cellulose, which is important for the hemostatic effect, was much lower in HA/OX compared to HA/CMC. Therefore, we decided to add etamsylate into the HA/CMC textile only. The aim was to verify if the addition of the active pharmaceutical ingredient would be therapeutically beneficial. There was much experimentation regarding the application of the etamsylate; however, only one method was effective enough to achieve the desirable amount of etamsylate on the nonwoven textile. Spraying a saturated solution of etamsylate onto the HA/CMC fabric has been shown to be suitable. Etamsylate in the amount of 15 mg/mL in 2-propanol was sprayed onto the textile, which was attached to a rotating collector designed for electrospinning. The solution was applied using a nozzle directed horizontally to the center, and vertically to the bottom third of the collector. The nozzle was placed 20 cm from the collector, which rotated at a set speed of 50 rpm. The nozzle position was specifically adjusted to achieve a high coating efficiency and a homogenous distribution of the active pharmaceutical ingredient. After spraying, the textile was left on the rotating collector until dry. Finally, an amount of 50 mL was sprayed on the nonwoven textile for in vivo testing. The total amount of the applied etamsylate was determined spectrophotometrically. The chosen detection method was based on the oxidation of etamsylate with ferric chloride and subsequent chelation using 1,10-phenanthroline [37]. The absorption maximum was observed at 510 nm. Calibration was determined in the concentration range 0.5–10 μg·mL^−1^ with a pre-built calibration curve y = 0.0988x − 0.0095, R^2^ = 0.9997. The final product for the in vivo experiment contained 4.80 g·m^−2^ of etamsylate.

### 2.4. Animal Model

Prior to in vivo testing, all material was examined by the internal control analytical laboratory for viability, sterilization, bacterial endotoxins, bioload, and residue. A MAT (Monocyte Activation Test) for pyrogen detection, was also provided. The work described was carried out in accordance with all the required guidelines for animal experiments, including the ARRIVE guidelines and EU Directive 2010/63/EU for animal experiments. From the beginning, the Scientific Committee for the Protection of Animals at the university, along with the Ministry of Agriculture, approved the in vivo concept (approval number 19-2016) according to the submitted experimental protocols. A partial nephrectomy model in rats was then performed [38]. Forty male Wistar laboratory rats (AnLab, Prague, Czech Republic) with an average weight of 244 ± 27 g were randomly assigned to 4 test groups of 10 rats. After acclimatization, animals were anesthetized using a mixture of tiletamine and zolazepam (Zoletil 100, Virbac S A., Carros, France). A dose of 65 mg/kg of body weight was applied intramuscularly. Once the animal was completely anesthetized, the partial nephrectomy of left kidney’s caudal pole was carried out immediately after the surgical opening of peritoneum. The open, bleeding wound was then covered with 1-cm^2^ pieces of the test hemostatic agents. Time to hemostasis was recorded. After hemostasis was achieved, the kidney was placed back into its original position in retroperitoneum, and the peritoneum and skin were sutured. The hemostatic material was left adhered to the wound. Half of the rats in the group were euthanized 3 days after the surgery by inhalational anesthesia isoflurane (Forane^®^, Aesica Queenborough Ltd., Queenborough, Kent, UK). The remaining rats were similarly euthanized after 30 days. The subsequent necropsy revealed the state of the peritoneal cavity. A section of the examined kidney was taken for subsequent histopathological assessment, immunohistochemical determination, and gene expression analysis. The isolated tissue samples were fixed in formaldehyde, embedded into paraffin and then stained for histopathological and immunohistochemical analyses.

### 2.5. Morphology of Fabrics

The textile samples were studied using a scanning electron microscope, SEM Ultra Plus (Zeiss, Oberkochen, Germany), with an acceleration mode of 3.5 kV. The dry textile was covered by a gold/palladium layer and analyzed using a secondary emission detector.

### 2.6. Measurement of Time to Hemostasis

Pieces of the materials measuring 1 cm^2^ were attached to the surgically created wounds in all tested groups. Time to complete hemostasis was measured in seconds.

### 2.7. Gene Expression (qRT-PCR) Analysis

RNA from the kidneys was isolated after mincing the kidney tissue with a sterile scalpel and its subsequent homogenization in 2-mL microtubes with 350 µl of RNAzol^®^ RT (Sigma Aldrich) and steel beads using TissueLyser II (QIAGEN, Hilden, Germany) for 10 min/30 Hz. The homogenized samples were then centrifuged at 12000 rpm for 15 min at 4 °C. For RNA isolation, supernatants with RNEasy Mini (QIAGEN) were used according to the manufacturer’s protocol. The RNA concentration was measured by NanoDrop™ One (Thermo Fisher Scientific); and cDNA synthesis was performed using High Capacity Reverse Transcription (Thermo Fisher Scientific) according to the manufacturer’s protocol. In brief, 1 μg of RNA was diluted in 14.2 μL of water and mixed with 2 μL of 10× RT Buffer, 0.8 μL of 100 mM dNTP, 2 μL of 10× RT Random Primers and 1 μL of MultiScribe Reverse Transcriptase. After a thorough mixing, samples were incubated in a thermocycler (Gene-Pro, Bioer Technology, Hangzhou, China) at these temperatures: 10 min at 25 °C, 120 min at 37 °C, 5 min at 85 °C, then at 4 °C until the end of the incubation. The gene expression was evaluated from 50-fold, water-diluted cDNA. The reaction mixture was prepared from 10 μL of Taq Man Fast Universal/Advanced master mix 2×, 1 μL of Taq Man Gene Expression Assay, 7 μL of water and 2 μL of diluted cDNA and incubated in StepOne (Thermo Fisher Scientific) for 20 s at 95 °C and then for 40 cycles at 95 °C (1 s) and at 60 °C (for 20 s). The relative gene expression was calculated using the 2^−ΔΔCT^ method [39]. Cytokines transforming growth factor beta (TGF-β) for fibrosis detection and an inflammation marker tumor necrosis factor alpha (TNF-α) were observed. RPL13A was used for housekeeping control.

### 2.8. Immunohistochemical Method

The release of the proinflammatory cytokine TNF-α and the TGF-β cytokine for fibrosis detection was detected in the immunohistochemical analysis. The prepared slides with tissue samples were immunostained using BenchMark^®^ ULTRA-Ventana Medical Systems (Roche, Basel, Switzerland) according to the immunostaining protocols [40]. The cytokines expression level was evaluated in five samples from each group using the following scale: no expression (−); light expression, less than 25% of positive cells (+); medium expression, 25%–50% of positive cells (++); very significant expression, more than 50% of positive cells (+++).

### 2.9. Histopathological Method

Part of the isolated tissue was histologically examined using hematoxylin and eosin (H&E) staining. Several parameters were assessed during the analysis: parameter D—destruction of the surrounding tissue, mainly destructive bleeding and necrosis; parameter F—inadequate extent of fibroproduction; parameter R—other reactions from the surrounding tissue, particularly dystrophic changes and the presence of protein in renal tubules; parameter I—inadequate extent of inflammatory infiltration. Each parameter was evaluated using the following scale: 0 (insignificant/light presence of the observed parameter), 1 (medium presence of the observed parameter), 2 (very significant presence of the observed parameter). The values detected in all the animals from one group were added together and a total destruction score was calculated.

### 2.10. Statistical Analysis

Data with homogenous variances were subjected to one-way ANOVA. Hemostatic data with heterogeneous variances were subjected to a nonparametric Kruskal–Wallis ANOVA for multiple comparisons using Statistica 10 software (StatSoft, Zličín, Czech Republic). Statistically significant differences were compared between all the analyzed groups. Data was expressed as means ± standard deviations.

## 3. Results

### 3.1. Morphology of Fabrics Evaluation

Morphology of all hemostatic materials was studied using a scanning electron microscope (SEM Ultra Plus, Zeiss, Oberkochen, Germany), because the structure has a significant influence on the functional properties of the hemostat. The images of all the tested materials are shown in Figure 2. Images did reveal structural differences in the nonwoven textiles. In HA/CMC—image B, clear fibers can be seen as evidence that the solution was properly utilized during the wet-spinning. The application of etamsylate, HA/CMC/ET—image D, (final addition of 4.8 g·m^−2^) changed the composition of the HA/CMC substrate (areal weight 52 g·m^−2^ without etamsylate) into a more compact-filled network. An imperfect net of fibers was observed when using the combination of hyaluronic acid with calcium salt of oxidized cellulose, HA/OX—image C, (areal weight 51 g·m^−2^), due to the cellulose’s insolubility in water and insufficient fiber formation. The reference ORC material—mage A (areal weight 37 g·m^−2^) had characteristic fibrillar morphology.

### 3.2. Time to Hemostasis Evaluation

All animals survived the surgical intervention without any postoperative complications. In each case, the bleeding was entirely stopped by the applied hemostat before the peritoneum was closed. A statistically significant shorter time to hemostasis completion in seconds was recorded in HA/CMC (19.2 s ± 11.8, *p* < 0.01) and in HA/CMC/ET (23.3 s ± 9.1, *p* < 0.05) in comparison with the new oxycellulose material HA/OX (41.8 s ± 16.2). Another statistical significance was recorded again in HA/CMC (*p* < 0.05) in comparison with the reference sample ORC (34.6 s ± 9.5). The obtained results are presented in the graph form in Figure 2.

### 3.3. Necropsy Report

Animals treated with the identical applied material were divided into two groups. The condition of the affected site was evaluated 3 days after surgery in the first group of test rats, while the second group was evaluated after 30 days. Immediately after necropsy, tissue samples were taken from the observed site for the histopathological evaluation. Histological images can be seen in Section 3.6.

Observation of the ORC fibrillar hemostatic textile after 3 days revealed residue from the material with traces of blood stuck to the wound. The ORC textile was resorbed after 30 days. One kidney of pale appearance was detected as a consequence of extensive blood loss. In the other cases, the kidneys exhibited a healthy appearance.

Regarding the HA/OX textile, residual material was found only in one wound during the 3-day evaluation. The nonwoven textile was adhered to the wound and there was a slight trace of blood in the peritoneal cavity. In all other cases, the material was resorbed completely. There were two cases from each evaluation, where the kidney looked pale. We attribute this finding to a greater blood loss.

In the HA/CMC and HA/CMC/ET textiles, the material was still stuck to the wound in two test rats during the 3-day evaluation. In these instances, the site had traces of blood and in one case, the kidney demonstrated a delicate structure. In the other evaluation, bioresorbability was proven and the kidney was characterized by healthy appearance in each test rat.

### 3.4. Gene Expression Evaluation

Results of the gene expression analysis in Figure 3 show that the material did not change the expression of the selected gene, with the exception of TNF-α, which was significantly decreased (*p* < 0.05) in the HA/CMC and HA/CMC/ET materials.

### 3.5. Immunohistochemical Evaluation

Two aforementioned cytokines TNF-α and TGF-β were also assessed during immunohistochemical analysis. The rate of expression is shown in the microscope images (Figure 4 and Figure 5) of the tested materials from the 3-day and 30-day assessment. Lower values of TNF-α were observed in the HA/OX material after 3 days, compared to a higher expression detected in the HA/CMC material and in the reference group. There was no difference between the tested groups in the 30-day evaluation, in respect to this cytokine. A higher expression of TGF-β in the 3-day evaluation was also detected in HA/CMC and ORC. A significant expression also persisted in the ORC samples in the 30-day evaluation. In contrast, a decrease in the expression in the new materials was observed.

### 3.6. Histopathological Evaluation

All the monitored parameters from the histopathological evaluation are shown in Table 1.

In the ORC textile, tissue samples taken 3 days after surgery and were characterized by necrosis and dispersed inflammatory cells, with a total destruction score of 25 (Figure 6A) After this period, the material residue irritated the tissue and at the time of the 30-day evaluation, the total score was increased to 29. Granulomatous reaction and chronic inflammatory infiltration at the site of the previous damage was clearly visible (Figure 6B)

In the HA/OX, tissue samples exhibited signs of the formation of excessive vascular granulation tissue after 3 days, with a destruction score of 35 (Figure 6C). In contrast, after 30 days, this agent demonstrated very visible regenerative and reparative processes (Figure 6D). The total destruction score was 20.

In the HA/CMC, the examined samples revealed necrosis and the presence of tubular protein cylinders together with hemorrhage and inflammatory infiltration (Figure 6E). The destruction score of this group was 29 after 3 days. This score decreased after 30 days and the total number in the end was 25. After a longer period, the parenchyma still displayed damaged tissue with granuloma and lingering inflammation (Figure 6F).

In the HA/CMC/ET, tissue samples were characterized by the presence of hematoma after 3 days (Figure 6G). The total destruction score was 29. A favorable healing reaction was observed after 30 days (Figure 6H), similar to results of the HA/OX at the time of the 30-day evaluation. The wound healed with a fibrous scar and only a slight, persistent inflammatory reaction still present. The destruction score was 20. If we consider both destruction scores, i.e., the score after 3 days and after 30 days, respectively, the HA/CMC/ET material yielded the best result.

## 4. Discussion

The study represents a pilot project for introducing novel hemostatic materials. Completely new nonwoven textiles using hyaluronic acid with sodium salt of carboxymethyl cellulose (ratio at 1:3), and hyaluronic acid with calcium salt of oxidized cellulose (ratio at 1:1) were prepared as a part of the experiment. Since there is a strong connection between rapid hemostasis and required healing, another hemostatic substance was added with the goal of improving the overall hemostatic potential. Etamsylate is an active pharmaceutical ingredient often used in surgery, which has been successfully applied in the therapy and prevention of local and systemic bleeding [28]. The above-mentioned textile composed of hyaluronic acid and carboxymethyl cellulose served as a substrate for etamsylate.

From a historical point of view, a more desirable hemostatic effect from the oxidized cellulose-based materials was presumed. Nevertheless, oxycellulose, strongly rooted in hemostatic research, particularly in the form of calcium salt, did not confirm its reputation during the experiment. Materials based on carboxymethyl cellulose (HA/CMC, HA/CMC/ET) turned out to be more effective against hemorrhage. Carboxymethyl cellulose textile (HA/CMC) reached a significantly shorter time to hemostasis (*p* < 0.05) compared to the reference oxycellulose material (ORC). Preparations consisting of hyaluronic acid and cellulose and potentially with an active pharmaceutical ingredient for hemostatic application are unique in this category and cannot easily be compared with materials that have been used previously. Comparing the results of another cellulose hemostatic study utilizing the same in vivo model, it can be assumed that the new composite preparations proved to possess more powerful hemostatic properties [38]. The carboxymethyl cellulose textiles also showed significant effect in comparison with the oxycellulose version (HA/OX) of the new product (*p* < 0.01 for HA/CMC, *p* < 0.05 for HA/CMC/ET). Their strong hemostatic effect may be the combined result of the structure and physico-chemical properties of the cellulose. Water insolubility of the calcium oxycellulose powder limited the cellulose content in the new nonwoven textile.

Interestingly, the carboxymethyl cellulose textile enriched by etamsylate (HA/CMC/ET) did not achieve a shorter time to hemostasis than the carboxymethyl cellulose textile without the active substance (HA/CMC), and the difference in results was practically negligible. From a statistical perspective, the HA/CMC textile proved to be more efficient. Sodium salt of carboxymethyl cellulose ensured blood absorption and the material dissolved in the fluid afterwards. It gelled very easily. Carboxyl groups were dissociated by the influence of sodium salts and the free particles could react with components of the blood. It was reported also that carboxymethyl cellulose interacts with fibrin to stabilize the clot [22]. Previously published studies revealed the effect of etamsylate only in specific situations and the exact mechanics of its action remains unclear [41]. Several authors also highlight the effect of other active substances, such as tranexamic acid, in comparison with etamsylate [42].

It was observed that the HA/CMC textile became somewhat tacky after its contact with blood. The application of etamsylate to this substrate strengthened the entire structure. The textile became more user-friendly and in its reinforced compact state, irritated the wound less according to the destruction score from the histology analysis.

To sum up, a significantly shorter time to hemostasis in seconds of the newly developed carboxymethyl cellulose materials was recorded. Furthermore, all the textiles proved their bioresorbability. In most cases, new textiles were resorbed after 3 days. The bioresorbability of the reference material was proved at the time of the 30-day assessment. The potential bioresorbability represented another benefit of the tested material. Therefore, it may be left in the body after surgical intervention without necessitating its removal, contrary to some commonly used non-bioresorsbable hemostats [5].

To investigate local tissue changes after an acute hemorrhage treated by a hemostat, and the influence of the hemostat on tissue healing, regeneration and gene expression, both immunohistochemical analysis and histopathological analysis were performed. The immune response is accompanied by the release of the proinflammatory cytokine TNF-α and the cytokine TGF-β as a regulator of fibrosis after surgical intervention resulting in a loss of blood [43]. Immunohistochemical determination assessed the tissue in sections and gene expression analysis processed the entire removed tissue. This may cause different results. According to the gene expression, all tested materials did not show a significant effect on the kidney tissue and demonstrated a safe, non-immunogenic influence. Studying the levels of the two cytokines TNF-α and TGF-β in immunohistochemical analysis, higher values were expected during the acute phase in contrast to the 30-day evaluation. The new HA/CMC material supports this physiological deflection.

In this type of experiment, the histopathology evaluation is the most important of all post-surgical analysis. The total destruction score with the lowest value represents the best hemostatic and healing potential. The evaluation after 3 days mostly reflected hemostatic effectiveness. It was shown that rapid hemostasis can reduce the undesirable reaction of local tissue [44]. In this case, carboxymethyl cellulose materials and the reference material reached lower destruction scores. In contrast, the new oxycellulose material was less efficient and yielded a much higher destruction score, which corresponded with the time to hemostasis. The complex structure and the impact on the local tissue were reflected in the 30-day evaluation results. All the newly-developed textiles containing hyaluronic acid had more favorable results than the reference material. Hyaluronic acid probably played a crucial role in this process, together with the mechanical properties of the material. Hyaluronic acid contributed to a faster healing and regeneration process. It is well known that hyaluronic acid has an exceptional effect on the healing of wounds. Hyaluronic acid promotes the migration and differentiation of mesenchymal and epithelial cells. It specifically improves the deposition of collagen and supports the process of angiogenesis [45]. Considering the values of the destruction scores from the 3-day and 30-day evaluations, the HA/CMC/ET material provided the best results. Thus, the rapid hemostasis and favorable influence on the tissue after surgery were confirmed. The fibrous structure of the reference fabric probably irritated the wound by releasing fibers and a correlated higher destruction score was observed.

## 5. Conclusions

To conclude, three novel nonwoven textiles combined with oxidized cellulose and hyaluronic acid, carboxymethyl cellulose with hyaluronic acid, and carboxymethyl cellulose and hyaluronic acid with an addition of etamsylate as the active pharmaceutical ingredient, were introduced in the study and assessed in the rat partial nephrectomy model in comparison with a commercially available product based on oxidized regenerated cellulose. Newly prepared fabrics with the carboxymethyl cellulose content yielded favorable hemostatic activity, bioresorbability, and non-irritability, and when taking into account the hyaluronic acid content, also had a superior effect on the tissue healing and regeneration. The addition of the active substance etamsylate did not significantly improve the hemostatic properties at the first stage of hemostasis, but the strengthened substrate with etamsylate was more suitable for manipulation and, along with its decent hemostatic activity, the material showed a beneficial influence on the local tissue reaction.

## Figures and Tables

**Figure 1 materials-13-01627-f001:**
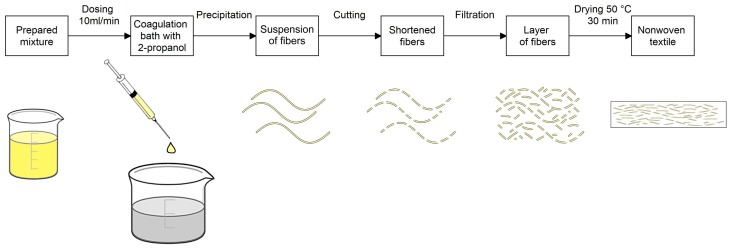
Schematic diagram illustrating the preparation of nonwoven textiles utilizing hyaluronic acid and cellulose.

**Figure 2 materials-13-01627-f002:**
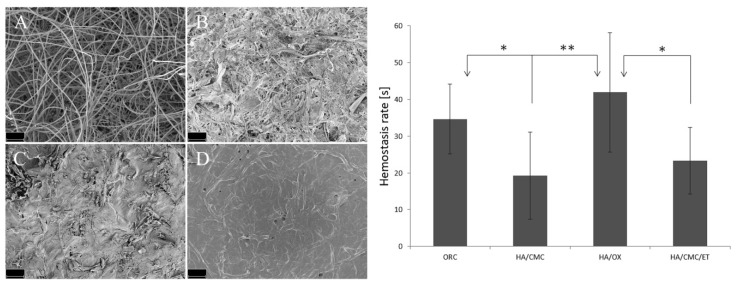
Left section—Morphology of all the hemostatic textiles under 100× magnification; scale bars represent 100 µm, (**A**) oxidized regenerated cellulose (ORC) textile, (**B**) hyaluronic acid and carboxymethyl cellulose (HA/CMC) textile, (**C**) hyaluronic acid and oxidized cellulose (HA/OX) textile, (**D**) HA/CMC/ET textile. Right section—time to hemostasis in seconds. Graph represents the means ± S.D. n = 10 for each study group. * *p* < 0.05, ** *p* < 0.01 in mutual comparison.

**Figure 3 materials-13-01627-f003:**
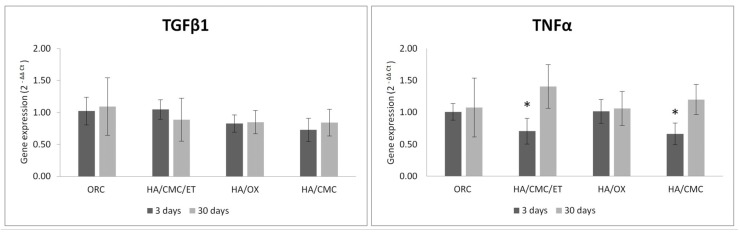
Results of gene expression analysis. Gene expression was compared to untreated controls. Graphs represent the means ± S.D. n = 4–5. * *p* < 0.05 in comparison to the control.

**Figure 4 materials-13-01627-f004:**
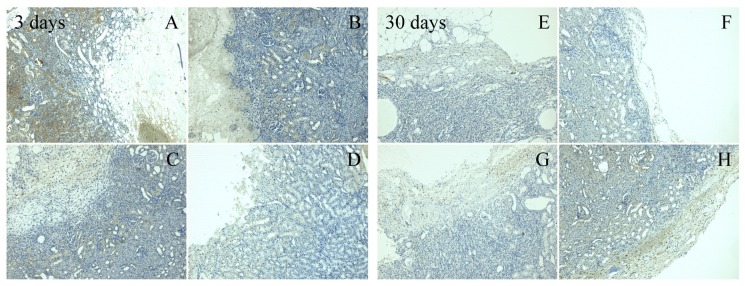
Immunohistochemical staining of TGF-β at 100× magnification. After 3 days (left section); (**A**) ORC, (**B**) HA/OX, (**C**) HA/CMC, (**D**) HA/CMC/ET: a very significant expression (+++) of the cytokine in all of the textiles **A**–**D**. After 30 days (right section); (**E**) ORC, (**F**) HA/OX, (**G**) HA/CMC, (**H**) HA/CMC/ET: medium expression (++) at **E**, only light expression (+) at **F**–**H**.

**Figure 5 materials-13-01627-f005:**
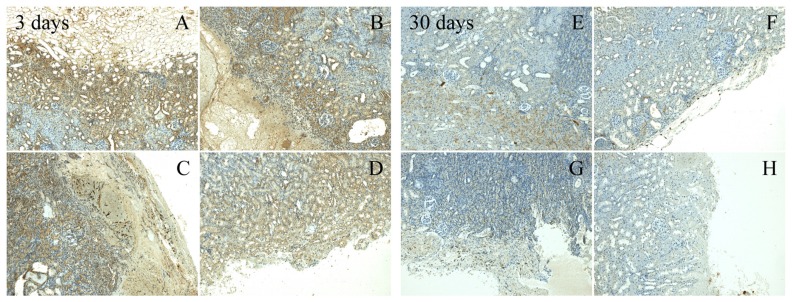
Immunohistochemical staining of TNf-α at 100× magnification. After 3 days (left section); (**A**) ORC, (**B**) HA/OX, (**C**) HA/CMC, (**D**) HA/CMC/ET: all new textiles **B**–**D** with light expression (+), reference textile **A**, medium expression (++). After 30 days (right section); (**E**) ORC, (**F**) HA/OX, (**G**) HA/CMC, (**H**) HA/CMC/ET: no expression (−) at **E**–**G**, light expression (+) at **H**.

**Figure 6 materials-13-01627-f006:**
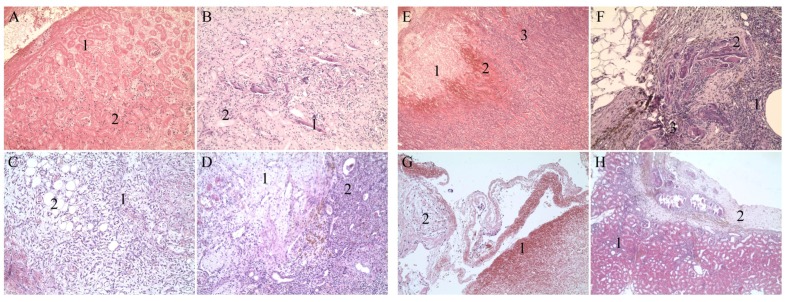
Histological images, H&E. Reference ORC textile: (**A**) Magnification: 100×; 3 days—necrotic tissue (1), inflammatory infiltrate (2). (**B**) Magnification 100×; 30 days—granuloma around the foreign material (1), chronic inflammatory infiltrate (2). HA/OX textile: (**C**) Magnification 100x; 3 days—vascular granulation tissue in the cutting line (1), adipose tissue (2). (**D**) Magnification 100x; 30 days—regular adequate scarring (1), traces of a delayed inflammation (2). HA/CMC textile: (**E**) Magnification 50×; 3 days—tissue necrosis (1), bleeding (2), protein cylinders (3). (**F**) Magnification 100×; 30 days—inflammation (1), granuloma (2), traces of a contamination (3). HA/CMC/ET textile: (**G**) Magnification 100×; 3 days—presence of a hematoma (1), damaged tissue in the cutting line (2). (**H**) Magnification 50×; 30 days—relatively favorable healing reaction with only temperate inflammation (1) and scarring (2).

**Table 1 materials-13-01627-t001:** Evaluated histopathological parameters.

Material	Resorbability	D	F	R	I	Destruction Score
ORC ^a^	no	2,1,2,2,1	1,1,1,1,1	1,1,1,1,2	1,1,1,1,2	25
HA/OX ^a^	yes	2,2,1,2,2	1,2,2,2,2	1,1,2,1,2	2,2,2,2,2	35
HA/CMC ^a^	yes	2,2,1,2,1	1,1,1,1,1	2,2,2,2,2	1,1,1,2,1	29
HA/CMC/ET ^a^	yes	1,2,1,2,2	1,2,0,2,2	1,1,2,1,2	1,2,0,2,2	29
ORC ^b^	yes	1,2,1,1,1	1,2,1,1,2	1,2,2,1,1	1,2,2,2,2	29
HA/OX ^b^	yes	1,0,0,1,1	1,1,1,0,1	1,1,1,2,1	2,0,1,2,2	20
HA/CMC ^b^	yes	1,1,1,2,1	1,1,1,1,1	1,1,2,2,1	2,1,1,2,1	25
HA/CMC/ET ^b^	yes	1,1,2,1,1	1,0,1,1,1	1,1,1,1,1	1,0,2,1,1	20

^a^ 3-day assessment, ^b^ 30-day assessment. D—tissue destruction, F—extent of fibroproduction, R—reaction of the surrounding tissue, I—inflammatory infiltration.

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
