# Peer review of "Composite Hemostatic Nonwoven Textiles Based on Hyaluronic Acid, Cellulose, and Etamsylate"

_materials, 2020, doi:10.3390/ma13071627_

Round 1
Reviewer 1 Report
The manuscript is focused on the application of hyaluronic acid, cellulose and etamsylate for composite hemostatic nonwoven textiles.
The topic is appropriate for the journal.
The title is adequate and correlate with the content of the article.
The abstract reports a consistent summary of the article findings, as well as the selected keywords.
The work has a clear structure.
All sections are properly written and required for a complete understanding.
Figures, tables and captions are correctly discussed.
The final summary match with the manuscript content presented and discussed.
Reviewer 2 Report
This is an interesting work by Suchý et al., introducing three novel nonwoven textiles and studying their hemostatic function in a rat partial nephrectomy model. I believe this is an impactful work that could potentially be of interest for a wide variety of research scientists in the field. There are some minor comments that I'd highly recommend to address:
1- The text requires major English editing to make sure that there is a smooth logical flow throughout the manuscript (there are multiple dictation and grammatical errors).
2- The Figure 1 could be significantly improved by some graphical presentation of each step, in form of schematics or actual photos of each experimental step. The current format is not informative at all.
3- Figure 2: report number of replicates (n) for each study group.
4- Section 3.3 of the Results has some important description of results from in vivo tests, but does not provide any figure/data to refer to. Some of this descriptive section does not seem to carry any scientific weight - e.g.: "With a producer guarantee of material’s bioresorbability after 7‐14 days". The main conclusions stated here would require some data (e.g. histology images) to back up.
5- Figure 4: panels cannot have identical labeling. The second set (day 30) should have E-H labels. Otherwise, this will be confusing.
6- Section 3.5: the comparisons made in terms of increased or decreased TNF‐alpha and TGF‐beta are not quite clear from the IHC images provided in Figs 4 and 5. A quantification of signal would be required in order to be able to draw such conclusions. For instance, just by skimming through Figure 4A through D, I cannot see any drastic difference is the signal.
7- I'd suggest to combine Figures 6 through 9 as they are all presenting same type of data, and putting them together in one figure will make this piece of data more clear and understandable.
Reviewer 3 Report
The paper describes the synthesis of three novel materials based on hyaluronic acid and oxidized cellulose for hemostasis. The hyaluronic acid and carboxymethyl cellulose showed the shortest hemostasis time. And all three materials showed good histological results and bioresorbaility after 3 days. The manuscript is well-written, and the data fully support the conclusions. I would suggest accepting in present form.
